# ON EXPLORING VISUAL ATTENTION SHRINKING FOR ACCELERATING VLMS FOR VIDEO UNDERSTANDING

## ABSTRACT

Vision-language models (VLMs) have shown promise in a variety of challenging video comprehension tasks. VLMs typically extract frames from the source video and take the corresponding visual tokens as input. However, a rapid increase in the number of visual tokens, e.g., when handling lengthy videos, can swiftly lead to a long-context dilemma for the inference efficiency of VLMs. Given that redundant and task-irrelevant information may exist in the visual tokens across both spatial and temporal axes, we advocate removing less important visual tokens during the prefilling stage of VLMs' inference procedure to improve the computation and storage efficiency. We first identify an interesting phenomenon termed as *Visual Attention Shrinking (VAS)*, wherein certain visual tokens receive progressively diminishing attention during the processing stages of the model. This implies that the model itself knows what to care about and what to discard. With this, we develop a robust algorithm to detect attention shrinking at each layer of the model using states from preceding layers. Based on the detection results, we perform token removal in both temporal and spatial axes. Our approach does not require parameterized modifications to the original VLM and is compatible with the prevalent KV cache strategy. Through extensive experiments across different VLMs, our approach witnesses an average speedup of $1.98\times$ in generating the first response token, utilizing only $47.2\%$ of the visual tokens, without significantly compromising the task performance. Additionally, when applied to the huge VILA1.5-40B, our method can achieve up to $4.16\times$ speedup compared to the vanilla model.

## 1 INTRODUCTION

Vision-language models (VLMs) excel at diverse visual understanding problems by integrating large language models (LLMs) with visual encoders (Liu et al., 2023a; Li et al., 2023a; Zhu et al., 2023; Ye et al., 2023; Bai et al., 2023; Wang et al., 2023). While early VLMs primarily focused on image-text tasks, recent models (Li et al., 2023b; Zhang et al., 2023; Lin et al., 2024a; Xu et al., 2024; Zhang et al., 2024a) have shifted the paradigm to video-text data. They usually extract frames from source videos and take the corresponding encoded visual tokens as input, exhibiting promise in diverse video understanding problems (Caba Heilbron et al., 2015; Li et al., 2024; Fu et al., 2024a).

Yet, existing video VLMs can encounter a significant efficiency dilemma when dealing with lengthy videos. In such scenarios, the number of visual tokens increases linearly with respect to the number of processed frames and can swiftly become the predominant cost factor during the prefilling stage of inference. Given the natural redundancy in visual signals, a reasonable solution is to reduce the number of visual tokens that need to be considered during the inference process. One can choose to merge the states within the visual encoders (Bolya et al., 2022; Haurum et al., 2023) so that the tokens fed to the VLMs decrease, but this is arguably suboptimal due to the loss of consideration of the textual information given to the VLMs. Similarly, some studies have attempted to adjust the architecture of the visual encoder (Shang et al., 2024; Arif et al., 2024), but this may introduce additional training requirements. Other works (Jin et al., 2024; Song et al., 2024; Li et al., 2025) have proposed new efficient VLMs by applying token merging strategies before the visual tokens enter the backbone of the language model, but these methods cannot be directly plug-and-play for other models. Recently, some works have directly explored the feasibility of token dropping during the inference stage of LLMs or VLMs (Xiao et al., 2023; Chen et al., 2024b; Lin et al., 2024b; Fu

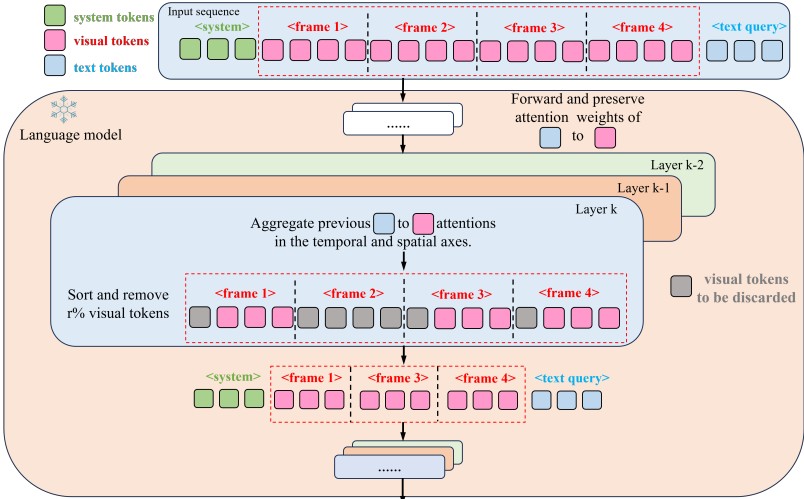

Figure 1: **Overview of our framework.** We do visual token reduction utilizing VAS during the prefilling stage of inference. It leverages the attention from text tokens to the visual tokens in previous layers, performing multiple visual token reductions to accelerate model inference.

et al., 2024b). Yet, they routinely focus on pure text or a single image combined with text, neglecting the spatial-temporal relationships between frames in video VLMs.

This paper explores eliminating the redundancy and task-irrelevant information in the visual tokens for video VLMs from both spatial and temporal axes. We first take a close inspection of the attention mechanism during the prefilling stage of the inference process of video VLMs, and observe an interesting phenomenon called *Visual Attention Shrinking (VAS)*—across the layers of the model, some frames or positions consistently exhibit a downward trend in attention scores. Given the computation mechanism of attention, we know such tokens are becoming less and less important for the generation of new tokens. So, it is a natural choice to discard them and let the model only care about the left ones. Based on this observation, we develop an algorithm that utilizes the attention states from previous layers to robustly detect attention shrinking, and continuously removes visual tokens based on the detection results. The overview of the proposed framework is illustrated in Figure 1.

We conduct extensive experiments on various VLMs, including PLLaVA (Xu et al., 2024), VILA (Lin et al., 2024a), and LongVA (Zhang et al., 2024a), across multiple video benchmarks to validate the effectiveness of our method. On average, our approach achieves a speedup of 1.98 times during the prefilling stage while utilizing less than half of the visual tokens, without compromising task performance. Additionally, when applied to the large VILA1.5-40B model, our method can achieve up to 4.16 times speedup compared to the vanilla model. We also conduct ablation studies to examine the impact of different settings on method.

## 2 RELATED WORK

### 2.1 VISION-LANGUAGE MODELS

Vision-Language Models are models designed to be capable of processing information from both visual and linguistic modalities. Early VLMs are likely to be lightweight, standalone models, such as Li et al. (2019), Kim et al. (2021). However, with the advent of large language models, VLMs begin to adopt these as their backbone and integrate relevant visual modules, producing numerous representative works. Alayrac et al. (2022) introduces modules such as the visual resampler and cross-attention adapter to achieve alignment between visual and textual inputs. Li et al. (2023a) employs Q-Former to integrate visual semantic vectors with language models. This kind of VLMs primarily focus on text-image inputs. But with the emergence of models like Video-LLaVA (Zhang et al., 2023), VLMs can process video inputs by sampling frames to acquire image lists and achieve

unified processing. Many researchers conduct extensive studies on the expansion of VLMs. Lin et al. (2024a) enhances the performance of VLMs by using interleaved training data. Xu et al. (2024) introduces a pooling strategy to adapt image-language pre-training models for video understanding tasks. Zhang et al. (2024a) extends the context length of language models, enabling VLMs to understand longer video sequences.

## 2.2 EFFICIENT MULTIMODAL LARGE LANGUAGE MODELS

Multimodal large language models have demonstrated excellent performance across various tasks. However, their large scale and high training and inference costs hinder their broader application. Therefore, research into efficient and lightweight models holds significant potential and meaning. Some works focus on optimizing the architecture. SPHINX-X (Gao et al., 2024) improves training efficiency by removing redundant visual encoders and skipping fully filled sub-images. Honeybee (Cha et al., 2024) introduces an innovative projector design that provides flexibility and local enhancement effects, achieving greater efficiency. Some researchers propose more efficient models (Jin et al., 2024; Song et al., 2024; Li et al., 2025) by leveraging token merging. Additionally, Some try to utilize small language models with excellent capabilities as their core components, such as MobileVLM (Chu et al., 2023), LLaVA-Phi (Zhu et al., 2024), VL-Mamba (Qiao et al., 2024). Furthermore, specific efforts have been dedicated to achieving more efficient visual processing, including optimizing attention mechanisms (Rao et al., 2021; Li et al., 2022), pruning (Yue et al., 2021; Kong et al., 2022; Chavan et al., 2022; Yu & Xiang, 2023), quantization (Wang et al., 2022; Liu et al., 2023b), etc.

## 2.3 TOKEN REDUCTION

Token pruning and merging techniques have been widely studied across different domains to improve the efficiency of transformer models. For instance, in classification tasks, researchers have found that not all tokens are necessary for accurate predictions, which led to the exploration of token pruning methods to reduce unnecessary information. ToMe (Bolya et al., 2022) employs a binary soft-matching algorithm for merging redundant tokens using the cosine similarity between key values. Kim et al. (2024) combines the advantages of token pruning and token merging, proposing a token fusion method to enhance the computational efficiency of visual transformers. Jiang et al. (2023) implements a prompt compression method for long contexts by reducing tokens, significantly decreasing the computational cost of language models. More related research (Shi et al., 2023; Chen et al., 2023; Cao et al., 2024) adopt similar approaches, refining and filtering tokens to eliminate potential redundancy and enhance efficiency.

## 3 METHOD

This section first reviews how VLMs handle and respond to multimodal inputs, then elaborates on the visual attention shrinking phenomenon within the attention maps of video VLMs. Based on this, a visual token reduction algorithm is developed for the prefilling stage of inference for acceleration.

### 3.1 PRELIMINARIES

VLMs typically contain three components: a visual encoder for extracting visual features, a projector for aligning visual and textual information, and a backbone LLM responsible for context understanding and output generation (Liu et al., 2024; Li et al., 2023a; Zhu et al., 2023). As said, the input to VLMs typically consists of visual data and texts, denoted as $V$ and $T$, respectively. The final tokens consist of three parts: system tokens, visual tokens, and text tokens, represented as $X_s$, $X_v$, and $X_t$. Text inputs are directly converted into corresponding tokens through the language model's tokenizer, resulting in $X_s, X_t$ while visual inputs are processed through the visual encoder and projector to form the final input. We denote the functions of the respective components as $f_{ve}$, and $f_{proj}$. Thus, the visual tokens can be obtained by:

$$X_v = f_{proj}(f_{ve}(V)). \tag{1}$$

Subsequently, by combining the visual information with the textual information, we obtain the input $X$ for the language model:

$$X = \text{Concat}(X_s, X_v, X_t). \tag{2}$$

For video inputs, the model samples a certain number of frames, denoted as $F$. The actual visual input can be considered as a multi-image input. For a single frame, it is further divided into multiple distinct patches, recorded as $P$. Therefore, the visual input $V$ encompasses both temporal and spatial dimensions. Assuming that for a given model, the number of sampled frames and patches are denoted as $m$ and $k$ respectively, we have:

$$V = [F_1, F_2, \ldots, F_m], \quad F_i = [P_{i,1}, P_{i,2}, \ldots, P_{i,k}]. \tag{3}$$

After processing through the visual encoder and projector, $X_v$ retains a similar spatial-temporal structure, representing a concatenation of multiple frames. Each frame consists of $n$ tokens, with each token conveying information about the corresponding position of the patch in the frames.

The attention mechanism is widely used in models, where the input at each layer is transformed into the corresponding query, key, and value vectors through the projection matrices $W_q$, $W_k$, and $W_v$. The attention weights $A$ are computed as:

$$A = softmax(\frac{(XW_q) \cdot (XW_k)^T}{\sqrt{d}}). \tag{4}$$

Here, $d$ represents the dimension of input features. By visualizing the attention weights, people can obtain the corresponding attention map, which aids in understanding and analyzing which parts of the input the model focuses on during inference.

Assume the input length is $L$ when the model generates the first token. By aggregating the attention weights of the multi-head attention, there is a $[L, L]$ matrix. We denote each element with $\alpha_l^{i,j}$, which is non-negative and reflects the attention level of the $i$-th token towards the $j$-th token and $l$ represents the layer number.

As the sequence length increases, the computational cost of attention mechanisms grows significantly. For each token, the model must compute attention scores relative to every other token, resulting in an $O(L^2)$ complexity. This exponential increase in calculations can lead to substantial memory and processing overhead. This challenge is particularly common in VLMs, where the visual feature $X_v$ is represented by a fixed number of tokens. In video tasks, as the number of sampled frames increases, the corresponding number of visual tokens also grows linearly, further exacerbating the computational burden.

## 3.2 VAS PHENOMENON

We first aim to analyze the importance of tokens using the attention map. $\sum_{i=1}^{L} \alpha_l^{i,j}$ calculates the sum of the attention weights from other tokens to the $j$-th token, as defined and identified in other works(Ge et al., 2023; Zhang et al., 2024b). This value reflects the level of attention that the $j$-th token receives from other tokens, and those with larger values can be referred to as "heavy hitters". In our subsequent research process, we also consider this metric for visual tokens, a larger value indicates that the information contained is relatively more important.

**Determine the Importance of Visual Tokens.** We attempt to identify metrics for assessing the importance of visual tokens. When evaluating the importance of visual tokens, directly summing the attention from other tokens to one single visual token may be somewhat unfair. Due to the existence of masks, earlier visual tokens receive additional attention from later tokens. Therefore, we use the following value to determine visual tokens' importance:

$$v^j = \sum_i^L \alpha^{i,j} \quad , \quad i \in X_t, j \in X_v. \tag{5}$$

We calculate this value for each visual token to assess its level of attention and importance. It only considers the attention of text tokens on visual tokens. Since the importance of visual tokens should vary depending on the specific text query, only the relevant tokens are taken into account to adapt to different contexts.

This design offers additional benefits. For text tokens, each visual token is visible, which eliminates concerns about fairness. Furthermore, the computation is more lightweight, as the number of text tokens in video models is significantly lower than that of visual tokens.

**Aggregate Attention.** We aggregate attention along the dimensions of time and space. We reshape the attention level into a matrix $I$ of size $[m, n]$, where $m$ represents the number of frames and $n$ denotes the number of tokens within each frame. Each row corresponds to a frame, and each column represents the attention received at a specific position. Consequently, We can sum over the temporal and spatial dimensions to obtain $T$ and $S$.

To validate the effectiveness of these values, we conduct a small verification experiment: during inference, we precompute the frame attention levels from one layer and group the visual tokens by frame. We then retain only one frame from adjacent pairs and discard the other one, first keeping the higher attention level and then the lower. The results show that the former achieves a higher accuracy than the latter (46.0 vs. 40.5), indicating that token groups with higher attention levels contain relatively more important information.

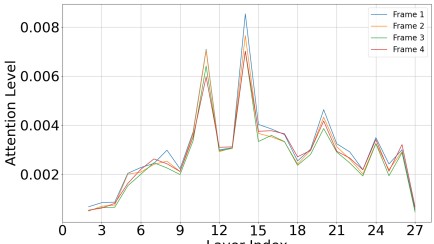 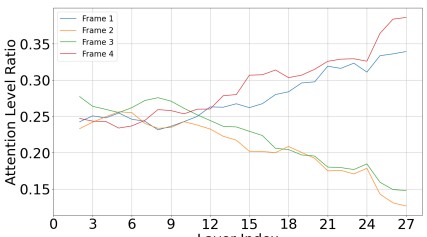

Figure 2: **Attention variation in temporal axis.** The left one illustrates the average variation curve of the attention levels across 4 frames in a subset of MVbench using PLLaVA-7B. The right one depicts the relative variation curve of the attention levels for a single sample. To make the curve of relative changes smoother, we compute the average of the attention levels of three adjacent layers to represent the value of the i-th layer.

**Attention Shrinking.** We then analyze the variation of the attention levels during the inference forward process. For specific models, they generally exhibit similar variation curves, as shown in Figure 2. We test on a minibatch using PLLaVA-7B model, sampling 16 frames. The figure shows the attention level trends for four selected frames. Notably, here we use the average to control the range of the vertical axis for clarity.

The attention levels of token groups exhibit similar variation trends with a sufficient sample size. This phenomenon is likely related to the differing projection matrices $W_q$, $W_k$, $W_v$ trained at each layer, with some layers focusing more on visual tokens, while others prioritize system and text tokens.

However, We identify a phenomenon we term *Visual Attention Shrinking* by comparing the relative magnitudes between several frames within individual instance. Some frames exhibit a consistently rising trend in attention, while others display an overall decreasing trend. The model's attention tends to spontaneously concentrate on certain specific frames depending on the query, which can be considered as key frames for addressing the corresponding query. This phenomenon is similarly reflected in the spatial dimension.

## 3.3 TOKEN REDUCTION BASED ON VAS

Based on the observation, we develop an algorithm that tracks the attention levels of previous layers to filter and drop visual tokens during the prefilling stage of inference. This method reduces computational load and accelerates model inference.

**Track Attention Levels.** We temporarily store the corresponding attention levels of previous layers during inference, which amount to the attention of text tokens to visual tokens. These data can be processed in parallel to obtain $T$ and $S$. After passing through the $k$-th layer of the decoder, we can rank the importance of visual tokens in the form like $(T_k - T_{k-1} + T_{k-2})$. A smaller value indicates a lower attention level, which means lower importance and can be utilized for subsequent token reduction.

---

**Algorithm 1** Leverage VAS to do visual token reduction

---

1: **Input**: $X$: The input to the language model, consisting of $X_S, X_V, X_T$, $m$: the number of sampled frames, $n$: the number of tokens per frame, $r\%$: The ratio for reducing, $l_{\text{start}}$: the start layer, $l_{\text{end}}$: the end layer

//During the prefilling stage of inference
2: **For** $k = 0 : max\_layer\_idx$ **do**
3:    **If** i is not between $l_{\text{start}}$ and $l_{\text{end}}$ **then**
4:      $X \leftarrow decoder\_layer_k(X)$;
5:    **Else**
6:      //Calculate the attention levels for visual tokens, with a dimension of [m,n]
7:      $X, I_k \leftarrow decoder\_layer_k(X)$;
8:      $T_k \leftarrow \text{sum}(I_k, \dim = 1), S_k \leftarrow \text{sum}(I_k, \dim = 0)$;   //$T_k, S_k$ have $m, n$ elements respectively
9:      //Judge whether to perform token reduction
10:      **If** (the ratio of $T_{k-2} > T_{k-1} > T_k$ or $S_{k-2} > S_{k-1} > S_k$ exceeds $r\%$) or (several layers not dropping) **then**
11:        Sort the attention levels of frames or positions based on corresponding $(T_k - T_{k-1} + T_{k-2})$ or $(S_k - S_{k-1} + S_{k-2})$ and remove the tokens with smaller values. Record the new number of frames $m'$ and the number of tokens per frame $n'$
12:        $X \leftarrow reduction(X)$;
13:        $m \leftarrow m', n \leftarrow n'$;
14:      **end If**
15:    **end If**
16: **end For**

---

**Gradual Token Removal.** We adopt a gradual removal strategy while striving to preserve the structural integrity of the video frame sequence established during model training. In video tasks, the number of visual tokens is substantial, and prematurely removing a large number of visual tokens may lead to irreversible information loss. Each individual visual token corresponds to specific frame and patch within a temporal group and a spatial group. We track the attention level changes of each group over the past three layers to determine whether to do token reduction.

To ensure the frequency of token dropping, we establish two conditions to determine when to remove tokens, with each dropping removing $r\%$ of the current visual tokens. First, if it is recorded that no token removal operation has been implemented for several consecutive layers, a visual token reduction will be performed and this guarantees a minimum threshold for token removal, preventing the retention of all visual tokens across excessive layers. The second condition is based on observing a consistent decline in attention levels for a sufficient proportion of temporal or spatial groups.

**More Details.** It is important to note that our gradual token removal occurs only during the inference's prefilling stage, namely in the forward process of generating the first token. This process does not utilize the KV cache, but the dynamic cache records the corresponding KV cache values at each layer. These values can be reused in the generation of subsequent tokens, eliminating the need for recalculation and dropping, so it does not conflict with the KV cache.

We set a start layer and an end layer. The start layer is designed to prevent the too early loss of critical visual tokens, while the end layer is set due to the limited number of remaining visual tokens, making further removal less meaningful. In experiments, we generally set the start layer to 2 and the end layer to the maximum layer index$-4$. However, in some situations, confusions between visual and text tokens may lead to abnormal outputs which may be because the number of visual tokens is significantly lower than that during training. In these cases, we may postpone the start layer and remove the remaining visual tokens at the end layer. This process ensures that the model generates stable and effective outputs. The complete design is shown in Algorithm 1.

## 4 EXPERIMENTS

### 4.1 EXPERIMENTS SETUP

**Models.** We apply the VAS method to several distinct open-source video understanding models, including PLLaVA-7B (Xu et al., 2024), VILA1.5-8B (Lin et al., 2024a), and LongVA-7B (Zhang et al., 2024a). Each of these models uses a different backbone language model (Chiang et al., 2023;

Table 1: Results of our method and other baselines on different models. Higher values indicate better performance. MB, VM, EG, ML, MQ denote the MVBench, VideoMME, Egoschema, MLVU, MSVD-QA datasets respectively. "Avg"calculates the average accuracy across all datasets. "Failed" means that the model is essentially unable to generate meaningful responses.

| Method | TTFT($\times$) | Img-Memory | MB | VM | EG | ML | MQ | Avg |
|---|---|---|---|---|---|---|---|---|
| PLLaVA-7B | 1.00 | 100% | 46.5 | 43.1 | **39.2** | 48.5 | **75.8/4.0** | 50.6 |
| +ToMe | 1.48 | 69.9% | 38.0 | 43.0 | 22.7 | Failed | Failed | — |
| +VCC | 1.67 | 53.1% | 44.5 | 41.3 | 23.8 | 45.6 | Failed | — |
| +FastV | 1.46 | 53.1% | 46.6 | 43.2 | 28.4 | 48.5 | 72.1/3.8 | 47.8 |
| +VAS | 1.87 | 47.7% | **46.9** | **44.3** | 34.9 | **49.0** | 72.4/3.8 | 49.5 |
| VILA1.5-8B | 1.00 | 100% | **48.5** | 46.0 | 49.4 | **47.5** | **66.4/3.8** | 51.6 |
| +ToMe | 1.39 | 70.1% | 48.0 | 45.0 | 49.7 | 45.0 | 62.2/3.6 | 50.0 |
| +VCC | 1.74 | 53.1% | 48.3 | 45.6 | 49.6 | 46.6 | 63.0/3.6 | 50.6 |
| +FastV | 1.42 | 53.1% | 48.2 | 45.2 | 48.9 | 34.9 | 65.5/3.7 | 48.5 |
| +VAS | 1.96 | 48.5% | **48.5** | **46.1** | **49.8** | 46.9 | 65.7/3.7 | 51.4 |
| LongVA-7B | 1.00 | 100% | **50.7** | 52.2 | **44.0** | 56.9 | **61.7/3.4** | 53.1 |
| +ToMe | 1.61 | 65.5% | 50.5 | 51.6 | 43.8 | 55.8 | 55.2/3.1 | 51.4 |
| +VCC | 1.85 | 53.6% | 48.2 | 51.6 | 41.7 | 48.6 | 58.3/3.3 | 49.7 |
| +FastV | 1.33 | 53.6% | Failed | 29.8 | 20.7 | Failed | Failed | — |
| +VAS | 2.10 | 45.3% | 48.3 | 51.7 | 42.9 | 51.0 | 57.6/3.2 | 50.3 |

Dubey et al., 2024; Yang et al., 2024). The sampling frame rates and other relevant configurations adhere to the default settings used during the evaluation of each respective model.

**Evaluation Tasks.** We test our method on different video evaluations, including MVBench (Li et al., 2024), VideoMME (Fu et al., 2024a), Egoschema (Mangalam et al., 2023), MLVU (Zhou et al., 2024), and MSVD-QA (Xu et al., 2017). Some of the evaluations comprehensively assess the model's capabilities in temporal and spatial video understanding and reasoning, while others specifically focus on long video comprehension or summarizing the entire video content. Most evaluations are conducted in the form of visual question answering, covering a diverse range of content and video types.

**Baseline and Implementation Details.** We will adhere to the respective settings of the models for the corresponding task types if applicable. The sampling frame numbers for the models PLLaVA-7B, VILA1.5-8B, and LongVA-7B are 16, 8, and 32, respectively, with corresponding totals of image tokens being 2304, 4608, and 1568. Additionally, we will utilize lmms-eval (Bo Li* & Liu, 2024) for evaluation on available datasets, unless the model cannot respond effectively in the evaluation.

We select FastV (Chen et al., 2024b) as our main comparative method. Because it is also a method using attention mechanisms to reduce visual tokens in multimodal models. It is important to note that although FastV is an excellent work, it is originally applied to text-image multimodal models and does not utilize KV cache. While this method is now compatible with KV cache and applicable to video models, it still conflicts with commonly used attention types in video models, such as SDPA Attention and Flash Attention, which hinders its acceleration effectiveness. As a result, its speedup ratio may be lower than expected.

We also select Token Merging (ToMe) and Visual Context Compression (VCC) as methods for comparison. ToMe (Chen et al., 2024b) uses the cosine similarity of key values to merge similar visual tokens, performing this operation across multiple consecutive layers, with a fixed number of visual tokens removed at each layer. VCC (Chen et al., 2024a) employs an average pooling strategy. For both FastV and VCC, we remove 50% of the visual tokens after the second layer. For ToMe, we remove a fixed number of visual tokens from each of the first 24 layers, adjusting the number of tokens removed based on the total count of visual tokens, specifically 48, 36, and 120. Refer to Appendix B to learn more about differences between above methods.

Table 2: Results of applying VAS to VILA1.5-40B on VideoMME to test acceleration potential. VAS* adopts a more aggressive strategy and conducted visual token reduction earlier.

| VILA1.5-40B | TTFT($\times$) | Img-Memory | Short | Medium | Long | Overall |
|---|---|---|---|---|---|---|
| Base | 1.00 | 100% | 68.3 | 57.1 | 51.7 | 59.0 |
| VAS | 2.63 | 37.1% | 66.3 | 55.3 | 50.6 | 56.4 |
| VAS* | 4.16 | 26.8% | 61.2 | 53.1 | 46.8 | 53.7 |

## 4.2 EXPERIMENTS RESULTS

The main results are presented in Table 1. We report the accuracy on various tasks, as well as the corresponding visual token reduction and acceleration effects. In most cases, the bottleneck for the generation time of subsequent tokens, aside from the first token, is loading the KV cache as quickly as possible. Therefore, we follow other works reporting the Time To First Token (TTFT). Img-Memory represents the proportion of remaining visual tokens in the KV cache after reduction compared to the original amount. This allows the same model to occupy less GPU memory during inference.

### 4.2.1 ANALYSIS ON DIFFERENT TASKS

Our method achieves acceleration across most tasks in different models while maintaining over 90% performance. Also, our gradual dynamic token reduction strategy is more flexible in reducing the number of visual tokens, resulting in better overall acceleration and lower memory usage.

However, we find that some methods encounter challenges in certain tasks. For instance, in the MSVD-QA task, attention-based token reduction does not yield satisfactory results. One possible reason is that this task requires video captioning, where the text queries are generally short and generic—for example, a question like "What is the man doing?" has limited content words. As a result, the text tokens may not effectively direct attention to the crucial parts of the visual tokens which are needed to find the answer, leading to information loss.

### 4.2.2 ANALYSIS ON DIFFERENT MODELS

Different language model architectures also have a significant impact. The Llama3 model used in VILA1.5-8B demonstrates the strongest capabilities, maintaining high-quality outputs even after a substantial reduction of visual tokens across all tasks. The Vicuna model used in PLLaVA-7B is also commonly employed, with many multimodal models relying on it as a backbone and benefiting from relatively mature training strategies, performing well in most tasks.

Meanwhile, some methods perform poorly or even fail to work on some models, generating meaningless repetition or an immediate end to the response. There are several reasons for this. On one hand, some previous methods are tested on only one model and have not been verified on others. On the other hand, there are inherent biases in the models themselves. For example, when certain prompts are inputted, the model tends to generate fixed response options after visual tokens are reduced. As for the performance on the LongVA model, since LongVA focuses on enhancing the model's long-context capabilities through textual data during training and does not train on long video data, its performance tends to degrade when the number of visual tokens is significantly reduced in several tasks.

### 4.2.3 APPLICATION ON LARGER MODELS

We also test our method on the VILA1.5-40B model. Since our approach requires attention from previous layers, the computation in the initial layers cannot be reduced. However, when applying to larger models, the increased number of decoder layers allows us to pursue a higher acceleration limit. Our experimental results demonstrate that this method remains effective on larger models, achieving a 4.16 times speedup while maintaining over 90% performance, as shown in Table 2.

Table 3: Some phased token reduction strategies tested on VideoMME using the VILA 1.5-8B model. Except for VAS, other methods reduce half current visual tokens' quantity tokens at the 12th and 20th layers..

| Method | TTFT($\times$) | Img-Memory | Acc |
|---|---|---|---|
| VAS | 1.96 | 47.8% | 46.1 |
| Random | 1.22 | — | 8.7 |
| Spatial average | 1.22 | 60.0% | 45.9 |
| Temporal average | 1.22 | 60.0% | 45.4 |

Table 4: Comparisons of different settings for the reduction ratio and maximum dropping intervals. The results are obtained using the VILA 1.5-8B model tested on MVBench.

| ratio | interval | TTFT($\times$) | Img-Memory | Acc |
|---|---|---|---|---|
| 20% | 3 | 1.70 | 42.7% | 48.3 |
| 30% | 3 | 2.06 | 39.6% | 48.2 |
| 40% | 3 | 2.23 | 32.5% | 47.7 |
| 50% | 3 | 2.36 | 27.8% | 47.5 |
| 20% | 4 | 1.57 | 59.0% | 48.7 |
| 30% | 4 | 2.02 | 42.4% | 48.5 |
| 40% | 4 | 2.11 | 37.9% | 47.1 |
| 50% | 4 | 2.30 | 30.8% | 46.2 |

## 4.3 ABLATIONS

### 4.3.1 COMPARISON WITH SOME TOY METHODS

Our method utilizes multiple non-consecutive layers for token reduction, which is different from others. This may be a more reliable and effective strategy. To investigate this, we design some corresponding simple methods for comparison. As shown in Table 3, we apply average token merging on the VILA1.5-8B model to reduce the number of tokens to half of the original twice. The impact of it on performance is relatively slight. So, training an appropriate adaptor module during inference may also serve as a promising token reduction method.

### 4.3.2 REDUCTION RATE AND RATIO

In our experiments, a reduction ratio controls the proportion of visual tokens reduced at each time. A maximum interval is also set to prevent continuous skipping of dropping. These two values significantly influence the efficiency of our method. Therefore, we adjust these parameters and conducted corresponding ablation studies, with the results shown in the Table 4. From the results observed, the two factors are not independent. When a relatively loose interval is paired with a faster reduction ratio, the results tend to worsen. They need to be matched, and a balance must be struck between performance and acceleration. Generally speaking, setting the reduction ratio to a conservative 30% is a good choice.

## 5 CONCLUSIONS

In this paper, we address the issue of slow long-context reasoning in VLMs when dealing with video tasks. We analyze the attention maps of the corresponding visual tokens and observe a phenomenon of visual attention shrinking. Based on this observation, we propose an algorithm that dynamically removes visual tokens during the inference prefilling stage, without requiring training, to reduce computational overhead and accelerate the inference of video VLMs.

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

# A ADDITIONAL EXPERIMENTS

In addition to the datasets shown in Table 1, we also validate our method on LongVideoBench (Wu et al., 2024) and MovieChat-1K (Song et al., 2024). Changes are not made to the experimental settings., thus the speedup ratios are close to those in Table 1. The results are shown in Table 5.

Table 5: Results of our method and other baselines on long video benchmarks. LVB, MC denote the LongVideoBench validation set, MovieChat-1K respectively.

| Method | Img-Memory | LVB(Acc) | MC(GPT eval) |
|---|---|---|---|
| VILA1.5-8B | 100% | 47.1 | 40.0/2.9 |
| +ToMe | 70.1% | 46.3 | 42.3/2.9 |
| +VCC | 53.1% | 46.8 | 41.7/3.0 |
| +FastV | 53.1% | 46.8 | 39.7/2.9 |
| +VAS | 47.9% | 47.4 | 42.5/3.0 |
| LongVA-7B | 100% | 52.6 | 40.7/2.7 |
| +ToMe | 65.5% | 52.1 | 40.5/2.7 |
| +VCC | 53.6% | 52.1 | 39.9/2.7 |
| +VAS | 47.5% | 52.4 | 40.3/2.7 |

In some experiments in the main text, our method shows a higher drop in performance and may sometimes appear slightly worse that other method. This is due to our adoption of a more aggressive token removal strategy. In fact, we can make a trade-off, balancing between speedup and accuracy. For example, we conduct additional experiments on the MLVU dataset and apply a more conservative removal strategy. We can achieve a 1.59X speedup using 63.3% of the visual tokens, with an accuracy of 56.7%, which is close to the base's 56.9% and it is higher than ToMe's 55.8%, which uses 65.5% of the visual tokens to achieve a 1.61X speedup.

It is the same when we test on VILA 1.5-40B. Our intention is to validate that our method can achieve greater acceleration on models with more layers. Therefore, we focus on achieving a higher speed-up ratio. We can also apply a onservative strategy and compare it with the FastV method. The results are shown in Table 6. It is evident that our method is better while also being flexible.

Table 6: Results of applying VAS and FastV to VILA1.5-40B on VideoMME to demonstrate that our method can still ensure performance when not overly prioritizing speedup ratios

| VILA1.5-40B | TTFT($\times$) | Img-Memory | Short | Medium | Long | Overall |
|---|---|---|---|---|---|---|
| Base | 1.00 | 100% | 68.3 | 57.1 | 51.7 | 59.0 |
| FastV | 1.53 | 54.2% | 65.4 | 56.1 | 51.7 | 57.8 |
| VAS | 1.84 | 48.6% | 68.0 | 56.2 | 52.4 | 58.8 |

We also provide the visualization of text-to-visual attention maps w/o or w reduction for comparison, as shown in Figure 3 Since the total number of tokens can reach thousands with a video input, we aggregate the attention for visual tokens from the same frame. After we removing the potentially unimportant visual tokens, the attention distribution of the remaining tokens is largely unaffected.

# B COMPARISONS OF DIFFERENT METHODS

In this section, we will specifically highlight the differences between the various methods used in the experiments. Before that, we first outline the key features of our approach. Our VAS method only leverages the attention of text tokens on visual tokens. This choice is based on two reasons: first, we believe that the important visual information varies depending on the text query, and second, it allows for more lightweight computation. Our method involves multiple rounds of token reduction,

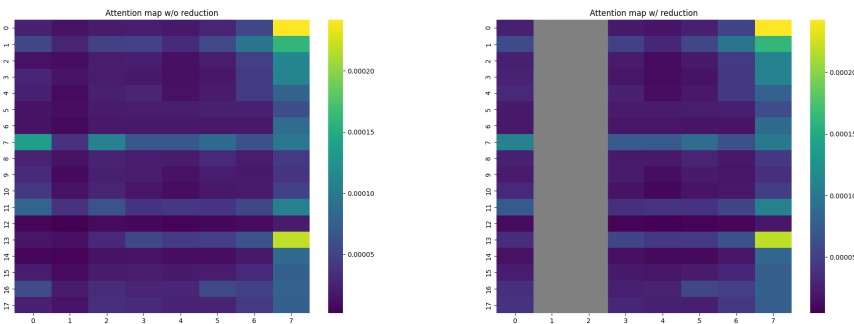

Figure 3: **Attention map w/o or w/ reduction.** The left one illustrates the text-to-visual attention without our VAS doing visual token reduction and the right one is with reduction. The gray areas represent the removed parts.

with $r\%$ of visual tokens removed in each round. As a result, the number of visual tokens used throughout the process gradually decreases.

FastV also utilizes the attention mechanism and incorporates a reduction ratio, typically set to $50\%$. Unlike our method, which performs multiple rounds of token reduction, FastV removes $50\%$ of the visual tokens in a single operation at a specific layer $k$, with the number of visual tokens remaining fixed after that. Generally, $k$ is chosen to be 2. The motivation behind FastV lies in the redundancy of visual tokens; however, it does not specifically consider the impact of different text queries. It uses global attention and removes the visual tokens that receive lower total attention. This approach may have limitations, as discussed in the main text.

VCC is straightforward, essentially performing an average pooling operation. In our implementation, we follow the settings used in FastV, reducing $50\%$ of the visual tokens after the second layer, with the number of tokens remaining unchanged thereafter. As a result, the memory usage of visual tokens in FastV and VCC is always the same in our experiments. This is a simple approach, though it is often not the optimal. Additionally, in video understanding model training, the image information of each frame has usually already undergone an average pooling operation.

ToMe is more complex. Unlike FastV or VCC, which perform token reduction once, it also differs from our flexible dynamic multi-round removal approach. First, ToMe uses the cosine similarity between the key values of visual tokens, rather than the attention scores. The idea behind this is that, for a given image, some patches may contain similar information, such as background color blocks. By identifying these similar color blocks and merging them in an average manner, ToMe reduces the number of visual tokens. Similar to VCC, it does not specifically account for the impact of different text queries. This process is applied in the initial several layers of the model, where a fixed number of visual tokens is reduced after each merge, rather than a percentage. The exact number of tokens reduced each layer is specified in the main text.

Anyway, these methods are initially tested on a single model or models with similar strong language model backbones, which may not be applicable in all scenarios. As a result, in our experiments, there are inevitably instances where these methods failed.

