# OpenReview forum: "On Exploring Visual Attention Shrinking for Accelerating VLMs for Video Understanding"
_ICLR.cc/2025/Conference — Submitted to ICLR 2025_

### Official Review · Reviewer_qUB7 · 2024-10-21

**Soundness:** 2
**Presentation:** 2
**Contribution:** 3
**Rating:** 6
**Confidence:** 5

**Summary:**

The large number of visual tokens creates significant inconvenience for both the training and inference of large multimodal models (LMMs). Therefore, reducing and removing unnecessary visual tokens is an important direction for development. This paper observes that certain visual tokens receive progressively less attention during the processing stages of the model. Based on this observation, token removal is performed along the temporal and spatial axes without requiring architectural modifications. This strategy can reduce the number of visual tokens by approximately 50%, with almost no performance degradation, while significantly speeding up inference.

**Strengths:**

1. The efficiency and performance of the proposed method are competitive compared to ToMe and FastV.

2. This is an architecture-agnostic approach, meaning it can be widely applied to various LMMs.

3. Although the pipeline figure in the paper is somewhat confusing, the textual organization is well-structured and makes it understandable.

4. In addition to the basic experiments, the authors included some observations (such as Figure 2) and ablation studies (Tables 3 and 4) to make the work more comprehensive.

5. The motivation of the paper is reasonable, and I believe this is indeed an urgent issue that needs to be addressed in LMMs.

**Weaknesses:**

1. The proposed VAS method does not actually show a sufficiently significant performance difference compared to ToMe [1] and FastV [2]. As shown in Figure 1, when VAS is applied to PLLaVA-7B, the performance on MVBench, VideoMME, and MLVU does not demonstrate a notable advantage (e.g., 46.6 vs 46.9, 48.5 vs 49.0). Furthermore, when VAS is applied to LongVA-7B, it consistently falls behind ToMe across all benchmarks.

2. There are many instances of "Failed" in Table 1, and the authors lack a reasonable explanation for this. The only mention of it is in Section 4.2.2, where it is briefly noted that "FastV’s aggressive reduction approach proves to be overly drastic, often resulting in model abnormal responses." However, this is merely a superficial description of the phenomenon.

3. The authors lack a direct comparison of the proposed VAS method with prior works, such as ToMe [1] and FastV [2], in terms of motivation and technique. Compared to FastV [2], both methods evaluate the importance of visual tokens based on attention weights, identifying tokens that can be discarded without sacrificing model performance. Both approaches use gradual token reduction strategies, ensuring that important tokens are preserved during the early stages of inference to prevent the loss of critical information.

4. The authors lack citations and discussions of other works focused on token merging, such as LlamaVid [3] and MovieChat [4], among others. Additionally, the authors' discussion of Chat-UniVi is inaccurate. The claims that it "attempted to adjust the architecture of the visual encoder" and "introduced additional training requirements" do not align with the original paper's descriptions.

5. Although the authors emphasize "lengthy videos" in the abstract, the experiments lack analysis and emphasis on VAS's performance and observations in long video sequences. For example, does attention dispersion become more severe with longer sequences? It would have been better if the authors had conducted more detailed observations on benchmarks like MovieChat-1K [4] or LongVideoBench [6].

[1] Bolya, Daniel, et al. "Token merging: Your vit but faster." arXiv preprint arXiv:2210.09461 (2022).

[2] Chen, Liang, et al. "An image is worth 1/2 tokens after layer 2: Plug-and-play inference acceleration for large vision-language models." arXiv preprint arXiv:2403.06764 (2024).

[3] Li, Yanwei, Chengyao Wang, and Jiaya Jia. "Llama-vid: An image is worth 2 tokens in large language models." European Conference on Computer Vision. Springer, Cham, 2025.

[4] Song, Enxin, et al. "Moviechat: From dense token to sparse memory for long video understanding." Proceedings of the IEEE/CVF Conference on Computer Vision and Pattern Recognition. 2024.

[5] Jin, Peng, et al. "Chat-univi: Unified visual representation empowers large language models with image and video understanding." Proceedings of the IEEE/CVF Conference on Computer Vision and Pattern Recognition. 2024.

[6] Wu, Haoning, et al. "Longvideobench: A benchmark for long-context interleaved video-language understanding." arXiv preprint arXiv:2407.15754 (2024).

**Questions:**

Please revise the Weaknesses section point by point. This is a paper with great potential. If the authors can provide additional responses to certain issues, discuss related work more thoroughly, and include more experiments and observations, I would be very happy to raise my score. Additionally, please redraw Figure 1, as it took me twice the time to understand what VAS is actually doing!

---

> ### Author Response · Authors · 2024-11-20
> **Response to Reviewer qUB7 (Part 1/2)**
>
> Thank you for taking the time to review our paper and providing insightful suggestions. We address your concerns point by point below.
>
> **W1:** Lack of sufficiently significant performance difference compared to other methods.
>
> It is worth noting that previous methods often test on a single model or models with similar strong language model backbones, which may not be applicable in all scenarios. However, our approach aims to generalize across different models and succeed. Additionally, since it is an untuned method, the performance improvements are limited. In the case of LongVA, the LongVA model primarily focuses on enhancing long-context capabilities through textual data during training and does not train on long video data, which may explain its poorer performance when the number of visual tokens is reduced significantly. This could also affect the ability of certain layers to extract key information through attention mechanisms. In contrast, ToMe does not utilize attention mechanisms but rather employs cosine similarity. In its application, we adopted a relatively conservative approach, retaining over 2000 visual tokens even in the final layer. Furthermore, ToMe evidently performs poorly on other models. Nevertheless, for fairness, we have included the corresponding results.
>
> We also conduct additional experiments with LongVA-7B, and apply a more conservative removal strategy on MLVU task, we achieve a 1.59X speedup using 63.3% of the visual tokens, with an accuracy of 56.7% (which is higher than ToMe's 55.8%, which using 65.5% of the visual tokens to achieve a 1.61X speedup). Just adjusting on this task, we can achieve an better average performance compared to ToMe. (51.6 vs 51.4)
>
> **W2:** Explanations for instances of "Failed" in Table 1.
>
> Related to the points mentioned in W1, previous methods often did not test on VLMs with different language model backbones, leading to failures in some scenarios. The most common examples of failure outputs are meaningless repetition or an immediate end of the response. The failure of FastV on LongVA is also related to the training strategy discussed in W1. LongVA primarily trains on textual data to enhance long-context capabilities. When half of the visual tokens are removed in early layers, the data diverges significantly from what was used during training, adversely affecting performance.
>
> **W3:** Comparisons of prior work.
>
> Thanks for your advice and we will supplement the relevant section later. It is worth to mention that FastV does not employ a gradual token reduction strategy. It removes R% of visual tokens at once at one selected layer. In contrast, ToMe gradually merges tokens using cosine similarity. Additionally, in practical applications, we only leverage the attention of textual information to visual information, while FastV utilizes global attention. When applying scaled dot-product attention or flash attention commonly used in video understanding models, global attention cannot be directed obtained, which is a significant drawback. In contrast to other methods, our method emphasizes retrieving and refining information from one modality based on another, and is applied specifically in video scenarios.

---

> ### Author Response · Authors · 2024-11-20
> **Response to Reviewer qUB7 (Part 2/2)**
>
> **W4:** Suggestions regarding related work.
>
> We greatly appreciate your suggestions regarding related work, as well as pointing out inaccuracies in our descriptions. These works focus more on proposing a new efficient LMM, rather than being a tuning-free and plug-and-play approach, which is different from ours. They achieve more efficient LMMs by performing token merging through lightweight operations before the visual information enters the large language model. We have similar ultimate goals, and thanks for your advice and we think these works to be inspiring and meaningful. We will cite your recommendations and make revisions in the relevant sections.
>
> **W5:** Analysis of long video sequences.
>
> In most current video understanding models, the final length of the video sequences is related to the number of sampled frames set by the model than to the actual length of the video itself. In our experiments, we observed that with a high number of sampled frames, the first and last frames tend to receive more attention. This may cause the model to focus more on the beginning and end of a long video, resulting in less accurate understanding of the content in the middle section. This may also explain why long video tasks are more challenging. In our work, we follow the model's default settings, where different models have different numbers of sampled frames and visual tokens. Additionally, we have conducted supplementary tests on the MovieChat-1K and LongVideoBench datasets as you suggested, and the results are as follows:
>
> | Method          | Img-Memory | LVB (Acc) | MC (GPT eval) |
> |------------------|------------|----------------------|-----------------------|
> | VILA 1.5-8B      | 100%       | 47.1                 | 40.0 / 2.9            |
> | + ToMe           | 70.1%      | 46.3                 | 42.3 / 2.9            |
> | + VCC            | 53.1%      | 46.8                 | 41.7 / 3.0            |
> | + FastV          | 53.1%      | 46.8                 | 39.7 / 2.9            |
> | + VAS            | 47.9%      | 47.4                 | 42.5 / 3.0            |
> | LongVA-7B       | 100%       | 52.6                 | 40.7 / 2.7            |
> | + ToMe           | 65.5%      | 52.1                 | 40.5 / 2.7            |
> | + VCC            | 53.6%      | 52.1                 | 39.9 / 2.7            |
> | + VAS            | 47.5%      | 52.4                 | 40.3 / 2.7            |
>
> Thanks for your constructive comments and positive feedback. Your deep understanding of the relevant field has highlighted several points we overlooked. We will need some time to revise and supplement the relevant sections of the paper based on your suggestions. Once again, thank you for taking the time to read our paper thoroughly and provide valuable suggestions.

---

> > ### Comment · Reviewer_qUB7 · 2024-11-21
> >
> > Thanks for your rebuttal work. Well done!
> >
> > For token merging, I believe doing in ViT will lead to better results (much better than you did in LLM layer, similar to [1]). It will be better if authors can make more clear clarify on exp setting.
> >
> > The extra exp results is good, and I believe the problem this paper wanna address is important. I'll raise from 5 to 6 once the updated manuscript is ready.
> >
> > [1] Kallini, Julie, et al. "MrT5: Dynamic Token Merging for Efficient Byte-level Language Models." arXiv preprint arXiv:2410.20771 (2024).

---

> > > ### Author Response · Authors · 2024-11-22
> > > **Acknowledgement**
> > >
> > > Thanks for your suggestions. We have redrawn Figure 1 and updated the manuscript. We also look forward to advancements in this field. Thank you again for your positive feedback!

---

> > > > ### Comment · Reviewer_qUB7 · 2024-11-22
> > > >
> > > > No worry. practice makes perfect. i've updated the my rating. have a good day.

---

### Official Review · Reviewer_jJKn · 2024-10-28

**Soundness:** 3
**Presentation:** 3
**Contribution:** 3
**Rating:** 5
**Confidence:** 4

**Summary:**

This paper identifies an interesting phenomenon termed as Visual Attention Shrinking (VAS), wherein certain visual tokens receive progressively diminishing attention during the processing stages of the model, and develop a robust algorithm to detect attention shrinking at each layer of the model using states from preceding layers.

The contributions are summaried as follows:
1. observe an interesting phenomenon called Visual Attention Shrinking.
2.  develop an algorithm that utilizes the attention states from previous layers to robustly detect attention shrinking, and continuously removes visual tokens based on the detection results.
3. extensive experiments on various VLMs, across multiple video benchmarks, validate the effectiveness of propsoed method.

**Strengths:**

1. The phenomenon of Visual Attention Shrinking where some frames or positions consistently exhibit a downward trend in attention scores, is interesting.
2. In the current field of video VLM, a large number of visual tokens are commonly used to represent videos. It is very meaningful to explore token reduction in this field.
3. The experiments are comprehensive, including multiple Benchmarks and video VLM models, and the method shows a significant acceleration for the models.

**Weaknesses:**

1. In certain settings, the proposed method experiences a higher drop in performance, such as with LongVA-7B and VILA1.5-40B.
2. Although the article focuses on accelerating the inference of video VLM models, it lacks substantial design specific to video.

**Questions:**

1.  Is the shape of $T$  $m$ or $n$? If it is $m$, then in Line 8 of Algorithm 1, $T$ should be summed over dimension 1. It appears there is a conflict between Line 8 and Line 10 in Algorithm 1.
2. How about multi-turn conversations?

---

> ### Author Response · Authors · 2024-11-20
> **Response to Reviewer jJKn**
>
> Thank you for the time to review our work. We are glad you think our research meaningful and comprehensive. We address your concerns point by point below.
>
> **W1:** There is a higher drop in performance in some settings.
>
> It is worth noting that previous methods used in VLMs often test on one model or models with similar strong LLM backbones, which may not be applicable in all scenarios. Our method selects three video VLMs PLLaVA, VILA, LongVA, whose LLM backbone and model design are significant different. The good performance on the three VLMs demonstrates the generalization of our methods. However, other methods fail to work on at least one model.
>
> Moreover, we discuss the performance in certain settings. LongVA focuses on enhancing the model's long-context capabilities through textual data during training and does not train on long video data, which may explain its poorer performance when the number of visual tokens is reduced greatly. Our method is only slightly worse than the ToMe method, which is because our method has a higher removal ratio. We conduct additional experiments on MLVU dataset and apply a more conservative removal strategy, we achieve a 1.59X speedup using 63.3% of the visual tokens, with an accuracy of 56.7% (which is close to the base's 56.9% and higher than ToMe's 55.8%, which using 65.5% of the visual tokens to achieve a 1.61X speedup). Only with this, our method delivers the best overall performance.
>
> For VILA1.5-40B, our intention was to validate that our method can achieve greater acceleration on models with more layers, hence we adopted a more aggressive token reduction strategy to achieve a higher speed-up ratio. We can also make a trade-off, balancing between speedup and accuracy. We conduct additional experiments, applying a more conservative strategy on VILA1.5-40B and comparing it with the FastV method. The results are as follows. As seen, our method outperforms FastV and offers greater flexibility.
>
> | VILA1.5-40B       | TTFT (×) | Img-Memory | Short | Medium | Long | Overall |
> |---------------|----------|------------|-------|--------|------|---------|
> | Base          | 1.00     | 100%       | 68.3  | 57.1   | 51.7 | 59.0    |
> | FastV          | 1.53     | 54.2%      | 65.4  | 56.1   | 51.7 | 57.8    |
> | VAS         | 1.84     | 48.6%      | 68.0  | 56.2   | 52.4 | 58.8    |
>
> **W2:** Design specific to video.
>
> From a broader perspective, our approach utilizes the model's inherent capabilities and attention mechanisms to retrieve and refine information from one modality based on another, allowing us to extract more important content. However, this does not mean that there is no design specific to video. For example, considering both temporal and spatial dimensions. To process attention more efficiently and perform token reduction, we aggregate attention scores across both spatial and temporal dimensions. Additionally, we compare the relative importance between frames, which can not be considered in VLMs with one image input.
>
> **Q1:** Confusion regarding Algorithm 1.
>
> Your understanding is correct, the shape of T is m. Apologize for any confusion caused by mistakes in the pseudocode. We will review the paper again and make the necessary corrections and clarifications to avoid ambiguity. Thank you for your careful reading!
>
> **Q2:** About multi-turn conversations.
>
> There seems to be limited examination of the multi-turn conversational capabilities of video understanding models. If you are referring to single video input, as mentioned in Section 3.3, Line 302, our method is compatible with the KV cache, allowing it to be directly used in multi-turn conversations. If you are considering multiple video inputs, this appears to be infrequently explored before. In this case, we might need to maintain a cache to record some indexes of each video along with the corresponding text tokens. Alternatively, concatenating multiple videos into one might facilitate processing.
>
> Thank you again for your feedback. We hope our responses help clarify some of your concerns.

---

> > ### Comment · Reviewer_jJKn · 2024-11-25
> >
> > 1. For multi-turn conversations, it means engaging in multiple rounds of Q&A on a single video. Given that the Visual Attention Shrinking described in the paper is instruction-aware, this means that the tokens that undergo shrinking will differ between the first and subsequent Q&A sessions due to the variation in questions. Consequently, the proposed method may lose its intended acceleration benefit in the context of multi-turn dialogues.
> >
> > 2. In Table 3, the 'Spatial average' toy method is only 0.2 points lower than 'VAS'. What would the corresponding results be if VAS were applied solely in the Spatial or Temporal dimension?

---

> ### Author Response · Authors · 2024-11-25
> **Response to Reviewer jJKn**
>
> We understand your concerns. To be precise, in multi-turn dialogues, when we choose to use the KV cache for subsequent rounds, the expected acceleration benefit will not be affected. In theory, to achieve the best performance, we should redo VAS at each Q&A round and select the important and relevant visual tokens, but doing so would impact the acceleration benefit. As mentioned in Section 4.2.1, for some descriptive tasks, text queries are generally short and generic, which can impact performance, but it doesn't mean it's unworkable. We can still retain relatively important visual tokens. Given that our goal is to achieve acceleration rather than pursuing higher performance. We are more inclined to use the KV cache in multi-turn dialogues.
>
> When we apply VAS solely in the spatial or temporal dimension, the results are 46.1 and 45.6 respectively. The accuracy is higher than that of the toy method, and the speedup ratios are slightly lower than the original VAS. In general, there is higher redundancy in the spatial dimension. It is worth noting that although the performance of 'Spatial average' is only slightly lower, its speedup ratio is only 1.22x, which is far inferior to the 1.96x achieved by VAS.
>
> We hope our reclarifications and explanations can resolve your concerns. If our response has addressed your concerns, we hope for your reconsideration in raising your score.
>
> If you have any additional questions or suggestions, we would be happy to have further discussions.

---

> > ### Comment · Reviewer_jJKn · 2024-12-02
> >
> > My primary concern is that the model's performance may significantly degrade if we opt for KV cache rather than recalculating VAS with each Q&A round. Could the authors please validate this concern using multi-turn dialogue benchmarks, such as MMDU[1] (even though it is not specifically a video benchmark)?
> >
> > [1] MMDU: A Multi-Turn Multi-Image Dialog Understanding Benchmark and Instruction-Tuning Dataset for LVLMs

---

> > > ### Author Response · Authors · 2024-12-02
> > >
> > > Due to time constraints and some issues encountered that could not be resolved immediately, we could only conduct evaluation on the MMDU to the best of our ability. Our method is designed for video understanding tasks. So when applied to image tasks, we make simplifications by not considering the temporal dimension. We use the VILA 1.5-8B model, but find that its 4K context window is insufficient. This limitation result in lower output quality in the later rounds of multi-turn dialogue, which significantly affect the evaluation results. Ultimately, we test on 10% of the samples, with the original overall score being 3.36 and the overall score after token removal being 3.28, which maintains performance above 95%.
> > >
> > > Of course, this is just a rough result. However, we believe that multi-turn dialogue scenarios are not the primary focus of our work. Currently, video understanding models are mainly applied in single-turn dialogue contexts, such as video captioning. We think that your main concern is whether retaining a fixed set of visual tokens is reliable when faced with different queries. This aspect can be partially addressed by previous works, such as references [1] and [2], which retain a fixed set of visual tokens with the same image inputs and have similar performance.
> > >
> > > On the other hand, we recently observe some researches exploring the mechanisms and information flow within VLMs, such as references [3] and [4]. According to their findings, high-level image information is compressed into the text embeddings in the early layers of VLMs. Our method takes a gradual token reduction strategy, and visual tokens are fully retained in the early several layers. Therefore, regardless of the final retention of visual tokens, our strategy accounts for the global visual information.
> > >
> > > Thank you for your feedback. We hope our clarifications and explanations are helpful to you.
> > >
> > > [1] Shang, Yuzhang, et al. "Llava-prumerge: Adaptive token reduction for efficient large multimodal models." arXiv preprint arXiv:2403.15388 (2024).
> > >
> > > [2] Arif, Kazi Hasan Ibn, et al. "HiRED: Attention-Guided Token Dropping for Efficient Inference of High-Resolution Vision-Language Models in Resource-Constrained Environments." arXiv preprint arXiv:2408.10945 (2024).
> > >
> > > [3] Zhang, Zhi, et al. "Cross-modal Information Flow in Multimodal Large Language Models." arXiv preprint arXiv:2411.18620 (2024).
> > >
> > > [4] Kaduri, Omri, Shai Bagon, and Tali Dekel. "What's in the Image? A Deep-Dive into the Vision of Vision Language Models." arXiv preprint arXiv:2411.17491 (2024).

---

> ### Author Response · Authors · 2024-11-30
>
> Dear reviewer,
>
> Thank you for your time in reviewing our work. We understand you may be quite busy rencently, but the revised deadline for the discussion is approaching. We kindly inquire whether our responses have addressed most of your concerns and hope you might reconsider adjusting your score. If there are any remaining issues, we would be happy to address them promptly.
>
> Best regards,
>
> The Authors

---

### Official Review · Reviewer_TQfB · 2024-11-03

**Soundness:** 3
**Presentation:** 3
**Contribution:** 2
**Rating:** 6
**Confidence:** 3

**Summary:**

This paper introduces a dynamic token removal method based on the phenomenon of Visual Attention Shrinking (VAS), which identifies less important visual tokens during inference. The algorithm operates without modifying the original VLM and is compatible with KV cache strategies.

**Strengths:**

The paper introduces the phenomenon of Visual Attention Shrinking (VAS) and develops a dynamic token removal algorithm based on this phenomenon, providing a new approach to improve the inference efficiency of video VLMs.
The proposed algorithm does not require parameterized modifications to the original VLM and is compatible with the prevalent KV cache strategy, demonstrating good generality.
The proposed method can perform token removal in both temporal and spatial dimensions, enhancing inference speed.

**Weaknesses:**

The paper primarily demonstrates the effectiveness of VAS through experimental results, but it lacks more intuitive visual analyses. It is recommended to use visualization tools (such as heatmaps, attention maps, etc.) to clearly illustrate the changes in attention distribution of the model before and after token removal, as well as the impact on the model's inference process.

The proposed method does not perform as well on certain specific tasks (such as Egoschema and MLVU) and fails to adequately explore the reasons for this.

**Questions:**

The paper mentions that the token reduction algorithm performs token reduction at certain layers, but what is the basis for selecting which layers to apply token reduction?
Can the dynamic visual token reduction algorithm you proposed be extended to other types of multimodal data beyond video?

---

> ### Author Response · Authors · 2024-11-20
> **Response to Reviewer TQfB**
>
> Thanks for your positive feedback and valuable suggestions! Below, we address the weaknesses and questions you proposed.
>
> **W:** Suggestions and Explanation for Performance on Some Datasets
>
> Thank you for your suggestion. We provide the visualization of text-to-visual attention maps w/ or w/o reduction for comparison in the Appendix. Since the total number of tokens can reach thousands with a video input, we aggregate the attention for visual tokens from the same frame during visualization. The gray areas represent the removed parts, and it can be seen that when we remove the potentially unimportant visual tokens, the attention distribution of the remaining tokens is largely unaffected.
>
> Regarding the poor performance on some tasks, we first discuss the Egoschema task. we think what you mean is that the performance on the PLLaVA-7B model drops significantly compared to the base model, but this does not affect our method is the best, as we still achieve an accuracy of 34.9%, while other methods do not exceed 30%. The underlying reason for this may be when the number of visual tokens is reduced, PLLaVA-7B tends to output fixed option under the default prompts. We tried to mitigate this issue, but it still hasn't been addressed completely. This may be due to the inherent bias the model has retained during training for specific inputs.
>
> For the MLVU task, it is inherently a challenging long-video dataset, and the main issue lies with the performance on the LongVA model. Our method is only slightly worse than the ToMe method, which is due to our adoption of a higher removal ratio. We conduct additional experiments, and when a more conservative removal strategy is applied, we achieve a 1.59X speedup using 63.3% of the visual tokens, with an accuracy of 56.7% (which is higher than ToMe's 55.8%, which using 65.5% of the visual tokens to achieve a 1.61X speedup). Furthermore, understanding long videos remains difficult for models, and LongVA focuses on enhancing the model's long-context capabilities through textual data during training and does not train on long video data, meaning that some layers' attention mechanisms cannot extract key information effectively. The FastV method, which also uses attention mechanisms, performs poorly and is even almost not applicable to it.
>
> **Q:** Some Application Details and Extensibility
>
> The relevant details are described in Section 3.3. In brief, we begin by applying token reduction to the current layer when we detect that a proportion of visual token attention has dropped in both the previous two layers. However, this may lead to infinite waiting, so we also record and apply token reduction if no token reduction has been made several layers.
>
> Essentially, our method functions like a retrieval process, where one modality's information is used to retrieve key content from another modality. The motivation here is to leverage the model's inherent ability and use the attention mechanism of textual informations to extract key visual information. Generally, models are most capable when handling textual information. Additionally, if some models have a unified representation across different modalities, this strategy can be extended to other modalities as well. How to process other modalities specifically can be explored in future work.
>
> Once again, thank you for your constructive feedback. It has provided us with valuable insights, and hope you are satisfied with our response.

---

> ### Author Response · Authors · 2024-11-30
>
> Dear reviewer,
>
> Thank you for your time in reviewing our work. We understand you may be quite busy rencently, but the revised deadline for the discussion is approaching. We kindly inquire whether our responses have addressed most of your concerns and look forward to your feedback. If there are any remaining issues, we would be happy to address them promptly.
>
> Best regards,
>
> The Authors

---

### Official Review · Reviewer_6r1M · 2024-11-03

**Soundness:** 2
**Presentation:** 3
**Contribution:** 2
**Rating:** 6
**Confidence:** 5

**Summary:**

This paper proposes a method to improve the efficiency of vision-language models for video tasks by reducing redundant visual tokens during inference. The authors introduce an algorithm to detect and remove low-importance tokens along spatial and temporal dimensions. Their approach requires no parameter changes, is compatible with KV caching, and achieves substantial speedups, using only 47.2% of tokens without degrading task performance, especially benefiting large models like VILA1.5-40B.

**Strengths:**

1. Tuning-free and plug-and-play: Being tuning-free and plug-and-play, The proposed method can be seamlessly integrated with existing VLMs without the need for extensive modifications or retraining, facilitating broader adoption.
2. Efficient Token Usage. Removing less important visual tokens during inference is an intuitive motivation.
3. Well-structured presentation. The presentation of this paper is clear and easy to understand.

**Weaknesses:**

1. Lack sufficient experimental support. It would be beneficial to include evaluations on other challenging video benchmarks, such as long video datasets, to validate effectiveness and enable a more comprehensive comparison.
2. Limited novelty. The concept of removing redundant tokens based on attention scores is not entirely new and has been explored in other VLM and transformer-based model optimizations. This approach may not offer a substantial advancement beyond existing methods.

Minor issues: These are some typos in the paper, such as Line 464 and the captions of Table 1.

**Questions:**

See weaknesses.

---

> ### Author Response · Authors · 2024-11-20
> **Response to Reviewer 6r1M**
>
> We sincerely thank you for taking the time to review our paper. We are glad you think our work can have a broad adoption. We address your concerns point by point below.
>
> **W1:** Lack of Sufficient Experimental Support.
>
> Thanks for your advice. The datasets we select in fact already include specialized long video datasets such as MLVU [1], in which the longest video lasts up to two hours. Additionally, we also use datasets like VideoMME [2], which include short, medium, and long videos spanning 6 primary visual domains with 30 subfields to ensure broad scenario generalizability. As shown in Table 1 of the main paper, our method outperforms other methods on these datasets. To further validate the effectiveness of our method, we conducted additional experiments on the LongVideoBench [3] and MovieChat-1K [4] datasets, and our method still works well. The results are as follows:
>
> | Method          | Img-Memory | LVB (Acc) | MC (GPT eval) |
> |------------------|------------|----------------------|-----------------------|
> | VILA 1.5-8B      | 100%       | 47.1                 | 40.0 / 2.9            |
> | + ToMe           | 70.1%      | 46.3                 | 42.3 / 2.9            |
> | + VCC            | 53.1%      | 46.8                 | 41.7 / 3.0            |
> | + FastV          | 53.1%      | 46.8                 | 39.7 / 2.9            |
> | + VAS            | 47.9%      | 47.4                 | 42.5 / 3.0            |
> | LongVA-7B       | 100%       | 52.6                 | 40.7 / 2.7            |
> | + ToMe           | 65.5%      | 52.1                 | 40.5 / 2.7            |
> | + VCC            | 53.6%      | 52.1                 | 39.9 / 2.7            |
> | + VAS            | 47.5%      | 52.4                 | 40.3 / 2.7            |
>
> **W2:** Limited Novelty.
>
> It is true that using attention scores for further analysis in models is a common practice. However, previous works like StreamingLLM [5] usually did not consider targeted design for inputs with both visual and textual information. Also, as we mention in the Introduction and Section 3.2, the important tokens should vary depending on the text query. Unlike the work of PruMerge [6] et al., we focus primarily on the attention from the text modality to the visual modality, as the visual information that needs to be emphasized should vary depending on the text query.
>
> Moreover, our method involves multiple dynamic pruning steps different from FastV [7]. The underlying principle is to leverage the model's inherent reasoning ability during the forward process to extract key information. We specifically consider both the temporal and spatial dimensions for video data to achieve efficient removal, which was not considered in previous methods for image VLMs. These different designs can potentially be transferred to other modalities in future works.
>
> Additionally, we truly appreciate your careful reading of the paper. We have corrected relevant typos and will later conduct a thorough review of the entire paper to revise possible remaining issues. Thank you again for your helpful suggestions and assistance.
>
> [1] Zhou J, Shu Y, Zhao B, et al. MLVU: A Comprehensive Benchmark for Multi-Task Long Video Understanding. arXiv preprint arXiv:2406.04264, 2024.
>
> [2] Fu, Chaoyou, et al. "Video-mme: The first-ever comprehensive evaluation benchmark of multi-modal llms in video analysis." arXiv preprint arXiv:2405.21075 (2024).
>
> [3] Wu, Haoning, et al. "Longvideobench: A benchmark for long-context interleaved video-language understanding." arXiv preprint arXiv:2407.15754 (2024).
>
> [4] Song, Enxin, et al. "Moviechat: From dense token to sparse memory for long video understanding." Proceedings of the IEEE/CVF Conference on Computer Vision and Pattern Recognition. 2024.
>
> [5] Xiao, Guangxuan, et al. "Efficient streaming language models with attention sinks." arXiv preprint arXiv:2309.17453 (2023).
>
> [6] Shang, Yuzhang, et al. "Llava-prumerge: Adaptive token reduction for efficient large multimodal models." arXiv preprint arXiv:2403.15388 (2024).
>
> [7] Chen, Liang, et al. "An image is worth 1/2 tokens after layer 2: Plug-and-play inference acceleration for large vision-language models." European Conference on Computer Vision. Springer, Cham, 2025.

---

> > ### Author Response · Authors · 2024-11-30
> >
> > Dear reviewer,
> >
> > Thank you for your time in reviewing our work. We understand you may be quite busy rencently, but the revised deadline for the discussion is approaching. We kindly inquire whether our responses have addressed most of your concerns and look forward to your feedback. If there are any remaining issues, we would be happy to address them promptly.
> >
> > Best regards,
> >
> > The Authors

---

### Author Response · Authors · 2024-11-22
**General Response**

Dear reviewers,

We appreciate each reviewer for the time spent reviewing our paper. Thanks for your positive attitude towards our work.

Thanks for your valuable questions and constructive suggestions, which have inspired us to address some of the previous shortcomings. We have responded to your concerns and uploaded a revised version. Here, we would like to briefly outline the changes made:

+ We correct the inaccurate description in the Introduction and redraw Figure 1, also adding relevant work.

+ In the Experiments section, we clarify the differences between the various methods and explain the reasons for the failures of some methods.

+ Additional experimental results addressing the reviewers' concerns are included in the Appendix.

+ Minor adjustments, such as correcting mistakes and avoiding ambiguous phrasing.

If you are willing, you can take a look at the revised version, where we highlight the main modifications in red in the main text. Of course, I believe and hope that our responses will address most of your concerns.

Thank you again for your feedback and valuable suggestions.

Best regards,

The Authors

---

### Meta-Review · Area_Chair_iAYG · 2024-12-19

**Metareview:**

This paper proposed a novel method to reduce video tokens both spatially and temporally to improve the efficiency of VLMs for video understanding. The proposed method is based on an observation denoted as visual attention shrinking. Proposed method was tested on different VLMs across several benchmarks show its effectiveness.

As pointed out by several reviewers, the proposed method is tuning-free and plug-and-play and can efficiently reduce the number of video tokens both spatially and temporally, showing good potential. Reviewers also pointed out some concerns: 1. Limited novelty (reviewer 6r1M) and lack of special design for video (reviewer jJKn) . 2. Missing some implementation details (reviewer TQfB). Considering all discussions, the difference of the proposed method is not significant enough from existing ones and the missing of implementation details makes the proposed method ad-hoc. I recommend rejection.

**Additional Comments On Reviewer Discussion:**

Reviewer 6r1M pointed out the concerns about the novelty of this paper, while addressing these concerns, the authors mentioned that FastV is designed for image VLMs while the proposed method is for video which takes temporal token reduction into consideration. However, when addressing the concerns of reviewer jJKn, the authors mentioned that applying the proposed method to spatial only achieves the same performance with current one, this brings the concerns about whether it is necessary to apply VAS to temporal dimension and the novelty of the proposed method is thus limited.

Reviewer TQfB had some questions about the details about the basis for selecting which layers to apply token reduction. The authors referred to section 3.3 for implementation details. However, the description in section 3.3 is not accurate, how to choose start and end layer exactly for each model is missing. In line 10 of Algorithm 1, the condition "several layers not dropping" is too vague as a scientific paper.

The overall score of this paper is border line, and the authors addressed concerns raised by reviewer qUB7. After carefully considering reviewers' comments and the discussion, I acknowledge the potential of the paper, but believe there are still concerns need to be addressed.

---

### Decision · Program_Chairs · 2025-01-22

Reject